# A Learning Strategy for Contrast-agnostic MRI Segmentation

Benjamin Billot[1]                                          BENJAMIN.BILLOT.18@UCL.AC.UK
Douglas N. Greve[2]                                           DGREVE@MGH.HARVARD.EDU
Koen Van Leemput[2,3]                                                      KVLE@DTU.DK
Bruce Fischl[2,4,5]                                         FISCHL@NMR.MGH.HARVARD.EDU
Juan Eugenio Iglesias[*1,2,4]                                       E.IGLESIAS@UCL.AC.UK
Adrian V. Dalca[*2,4]                                                   ADALCA@MIT.EDU

[1] *Center for Medical Image Computing, University College London, UK*

[2] *Martinos Center for Biomedical Imaging, Massachusetts General Hospital and Harvard Medical School, USA*

[3] *Department of Health Technology, Technical University of Denmark, Denmark*

[4] *Computer Science and Artificial Intelligence Laboratory, Massachusetts Institute of Technology, USA*

[5] *Program in Health Sciences and Technology, Massachusetts Institute of Technology, USA*

## Abstract

We present a deep learning strategy for contrast-agnostic semantic segmentation of unpreprocessed brain MRI scans, without requiring additional training or fine-tuning for new modalities. Classical Bayesian methods address this segmentation problem with unsupervised intensity models, but require significant computational resources. In contrast, learning-based methods can be fast at test time, but are sensitive to the data available at training. Our proposed learning method, SynthSeg, leverages a set of training segmentations (no intensity images required) to generate synthetic scans of widely varying contrasts on the fly during training. These scans are produced using the generative model of the classical Bayesian segmentation framework, with randomly sampled parameters for appearance, deformation, noise, and bias field. Because each mini-batch has a different synthetic contrast, the final network is not biased towards any specific MRI contrast. We comprehensively evaluate our approach on four datasets comprising over 1,000 subjects and four MR contrasts. The results show that our approach successfully segments every contrast in the data, performing slightly better than classical Bayesian segmentation, and three orders of magnitude faster. Moreover, even within the same type of MRI contrast, our strategy generalizes significantly better across datasets, compared to training using real images. Finally, we find that synthesizing a broad range of contrasts, even if unrealistic, increases the generalization of the neural network. Our code and model are open source at https://github.com/BBillot/SynthSeg.

**Keywords:** segmentation, contrast agnostic, CNN, brain, MRI, data augmentation

---

* Contributed equally

## 1. Introduction

Segmentation of brain MR scans is an important task in neuroimaging, as it is a primary step in a wide array of subsequent analyses such as volumetry, morphology, and connectivity studies. Despite the success of modern supervised segmentation methods, especially convolutional neural networks (CNN), their adoption in neuroimaging has been hindered by the high variety in MRI contrasts. These approaches often require a large set of manually segmented preprocessed images *for each* desired contrast. However, since manual segmentation is costly, such supervision is often not available. A straightforward solution, implemented by widespread neuroimaging packages like FreeSurfer (Fischl, 2012) or FSL (Jenkinson et al., 2012), is to require a 3D, T1-weighted scan for every subject, which is aggressively preprocessed, then used for segmentation purposes. However, such a requirement precludes analysis of datasets for which 3D T1 scans are not available.

Robustness to MRI contrast variations has classically been achieved with Bayesian methods. These approaches rely on a generative model of brain MRI scans, which combines an anatomical prior (a statistical atlas) and a likelihood distribution. The likelihood typically models the image intensities of different brain regions as a Gaussian mixture model (GMM), as well as artifacts such as bias field. Test scans are segmented by "inverting" this generative model using Bayesian inference. If the GMM parameters are independently derived from each test scan in an unsupervised fashion (Van Leemput et al., 1999; Zhang et al., 2001; Ashburner and Friston, 2005), this approach is fully adaptive to MRI contrast. In some cases, *a priori* information is included in the parameters, which constrains the method to a specific contrast (Wells et al., 1996; Fischl et al., 2002; Patenaude et al., 2011) – yet even these methods are generally robust to small contrast variations. Such robustness is an important reason why Bayesian techniques remain at the core of all major neuroimaging packages, such as FreeSurfer, FSL, or SPM (Ashburner, 2012). However, these strategies require significant computational resources (tens of minutes per scan) compared to deep learning methods, limiting large-scale deployment or time-sensitive applications.

Another popular family of neuroimaging segmentation methods is multi-atlas segmentation (MAS) (Rohlfing et al., 2004; Iglesias and Sabuncu, 2015). In MAS, several labeled scans ("atlases") are registered to the test scan, and their deformed labels are merged into a final segmentation with a label-fusion algorithm (Sabuncu et al., 2010). MAS was originally designed for intra-modality problems, but can be extended to cross-modality by using multi-modality registration metrics like mutual information (Wells et al., 1996; Maes et al., 1997). However, their performance in this scenario is poor, due to the limited accuracy of nonlinear registration algorithms across modalities (Iglesias et al., 2013). Another main drawback of MAS has traditionally been the high computational cost of the multiple nonlinear registrations. While this is quickly changing with the advent of fast, deep learning based registration techniques (Balakrishnan et al., 2019; de Vos et al., 2017), accurate deformable registration for arbitrary modalities has not been widely demonstrated with these methods.

The modern segmentation literature is dominated by CNNs (Milletari et al., 2016; Kamnitsas et al., 2017b), particularly the U-Net architecture (Ronneberger et al., 2015). Although CNNs produce fast and accurate segmentations when trained for modality-specific applications, they typically do not generalize well to image contrasts which are different from the training data (Akkus et al., 2017; Jog and Fischl, 2018; Karani et al., 2018). A possible

solution is to train a network with multi-modal data, possibly with modality dropout during training (Havaei et al., 2016), although this assumes access to manually labeled data on a wide range of acquisitions, which is problematic. One can also augment the training dataset with synthetic contrast variations that are not initially available from uni- or multi-modal scans (Chartsias et al., 2018; Huo et al., 2019; Kamnitsas et al., 2017a; Jog and Fischl, 2018). Recent papers have also shown that spatial and intensity data augmentation can improve network robustness (Chaitanya et al., 2019; Zhao et al., 2019). Although these approaches make segmentation CNNs adaptive to brain scans of observed contrasts, they remain limited to the modalities (real or simulated) present in the training data, and thus have reduced accuracy when tested on previously unseen MR contrasts.

To address modality-agnostic learning-based segmentation, a CNN was recently used to quickly solve the inference problem within the Bayesian segmentation framework (Dalca et al., 2019a). However, this method cannot be directly used to segment test scans of arbitrary contrasts, as it requires training on a set of unlabeled, preprocessed scans for each target modality.

We present SynthSeg, a novel learning strategy that enables automatic segmentation of *unpreprocessed* brain scans of *any* MRI contrast without any need for paired training data, re-training, or fine tuning. We train a CNN using a dataset of only segmentation maps: synthetic images are produced by sampling a generative model of Bayesian segmentation, conditioned on a segmentation map. By sampling model parameters randomly at every mini-batch, we expose the CNN to synthetic (and often unrealistic) contrasts during training, and force it to learn features that are inherently contrast agnostic. We demonstrate SynthSeg on four different MRI contrasts. We also show that SynthSeg generalizes across datasets of the same contrast better than a CNN trained on real images of this contrast from a specific dataset.

## 2. Methods

We first introduce the generative model for Bayesian MRI segmentation, and then describe our method, which builds on this framework to achieve modality-agnostic segmentation.

### 2.1. Classical generative model for Bayesian segmentation of brain MRI

The Bayesian segmentation framework relies on a probabilistic generative model for brain scans. Let $L$ be a 3D label (segmentation) map consisting of $J$ voxels, such that each voxel value $L_j$ is one of $K$ possible labels: $L_j \in \{1, \ldots, K\}$. The generative model starts with a prior anatomical distribution $p(L)$, typically represented as a (precomputed) statistical atlas $A$, which associates each voxel location with a $K$-length vector of label probabilities. Additionally, the atlas $A$ is endowed with a spatial deformation model: the label probabilities are warped with a field $\phi$, parameterized by $\theta_\phi$, which follows a distribution $p(\theta_\phi)$ chosen to encourage smooth deformations. The probability of observing $L$ is then:

$$p(L|A, \theta_\phi) = \prod_{j=1}^{J} [A \circ \phi(\theta_\phi)]_{j,L_j}, \tag{1}$$

where $[A \circ \phi(\theta_\phi)]_{j,L_j}$ is the probability of label $L_j$ given by the warped atlas at location $j$.

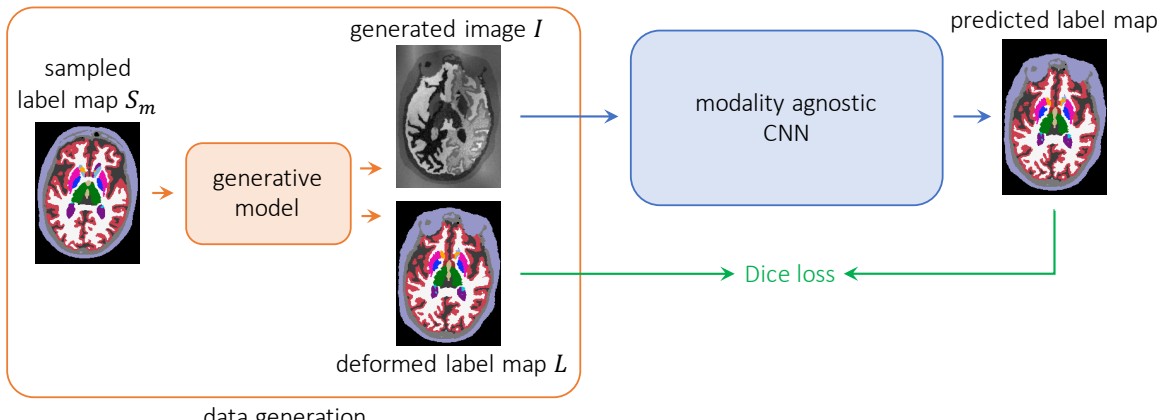

Figure 1: SynthSeg overview. The proposed data generation process selects one of the available label maps $S_m$ and employs a sampling strategy to synthesize an image-segmentation pair $\{I, L\}$, based on a well-established generative model of brain MRI. Specific generation steps are illustrated in Figure 2 and detailed in Algorithm 1. The pairs $\{I, L\}$ are used to train a CNN in a supervised fashion.

Given a label map $L$, the image likelihood $p(I|L)$ for its corresponding image $I$ is commonly modeled as a GMM (conditioned on $L$), modulated by smooth, multiplicative bias field noise (additive in the more convenient logarithmic domain). Specifically, each label $k \in \{1, ..., K\}$ is associated with a Gaussian distribution for intensities of mean $\mu_k$, and standard deviation $\sigma_k$. We group these Gaussian parameters into $\theta_G = \{\mu_1, \sigma_1, \ldots, \mu_K, \sigma_K\}$. The bias field is often modeled as a linear combination of smooth basis functions, where linear coefficients are grouped in $\theta_B$ (Larsen et al., 2014). The image likelihood is given by:

$$p(I|L, \theta_B, \theta_G) = \prod_j \mathcal{N}(I_j - B_j(\theta_B); \mu_{L_j}, \sigma^2_{L_j}), \tag{2}$$

where $\mathcal{N}(\cdot; \mu, \sigma^2)$ is the Gaussian distribution, $I_j$ is the image intensity at voxel $j$, and $B_j(\theta_B)$ is the bias field at voxel $j$. We assume that $I_j$ has been log-transformed, such that the bias field is additive, rather than multiplicative.

Bayesian segmentation uses Bayes's rule to "invert" this generative model to estimate $p(L|I)$, posing segmentation as an optimization problem. Such inversion often relies on computing point estimates for the model parameters. Fitting the Gaussian parameters $\theta_G$ to the intensity distribution of the test scan is what makes these methods contrast agnostic.

## 2.2. Proposed approach

We propose to train a segmentation CNN using synthetic data created on the fly with a generative model similar to that of Bayesian segmentation. Since the voxel independence assumption would yield extremely heterogeneous noisy images, we rely on a set of $M$ original label maps $S = \{S_m\}_{m=1}^M$ instead of random samples from a probabilistic atlas. We

---

**Algorithm 1:** Proposed Learning Strategy for SynthSeg

---

**Input:** $\{S_m\}_{m=1,\ldots,M}$            `// M segmentations`

**while** *not converged* **do**

    $i \sim \mathcal{U}_d(1, M)$               `// select input map`

    $\theta_{aff} \sim \mathcal{U}(a_{rot}, b_{rot}) \times \mathcal{U}(a_{sc}, b_{sc}) \times \mathcal{U}(a_{sh}, b_{sh}) \times \mathcal{U}(a_{tr}, b_{tr})$   `// affine parameters`

    $\theta_v \sim \mathcal{N}_{10 \times 10 \times 10 \times 3}(0, \sigma_{svf}^2)$         `// sample SVF parameters`

    $\phi_v(\theta_v) \leftarrow ScaleAndSquare[Upscale(\theta_v)]$     `// upscaling and integration`

    $\phi \leftarrow \phi_{aff}(\theta_{aff}) \circ \phi_v(\theta_v)$           `// form deformation`

    $L \leftarrow S_i \circ \phi$               `// deform selected label map`

    $(\mu_k, \sigma_k) \sim \mathcal{U}(a_\mu, b_\mu) \times \mathcal{U}(a_\sigma, b_\sigma), k = 1, \ldots, K$    `// sample Gaussian parameters`

    $G_j \sim \mathcal{N}(\mu_{L_{ij}}, \sigma_{L_{ij}})$          `// sample GMM image`

    $G^{blur} \leftarrow G * R(\sigma_{blur})$          `// Spatial blurring`

    $\theta_B \sim \mathcal{N}_{4 \times 4 \times 4}(0, \sigma_b^2)$        `// sample bias field parameters`

    $B \leftarrow \exp[Upscale(\theta_B)]$        `// upscaling and exponentiation`

    $G^{bias} \leftarrow G^{blur} \odot B$          `// bias field corruption`

    $\gamma \sim \mathcal{U}(a_\gamma, b_\gamma)$         `// gamma augmentation parameter`

    $I \leftarrow f(G^{bias}, \gamma)$      `// gamma and normalization via (4)`

    update CNN weights with pair $\{I, L\}$        `// SGD iteration`

**end**

---

also slightly blur the sampled intensities. The proposed learning strategy, detailed below, is summarized in Figure 1 and Algorithm 1, and exemplified in Figure 2.

**Data sampling:** In training, mini-batches are created by sampling image-segmentation pairs $\{I, L\}$ as follows. First, we randomly select a label map $S_i$ from the training dataset (Figure 2a), by sampling $i \sim \mathcal{U}_d(1, M)$, where $\mathcal{U}_d$ is the discrete uniform distribution.

Next, we generate a random deformation field $\phi$ to obtain a new anatomical map $L = S_i \circ \phi$. The deformation field $\phi$ is the composition of an affine and a deformable random transform, $\phi_{aff}$ and $\phi_v$, parameterized by $\theta_{aff}$ and $\theta_v$, respectively: $\theta_\phi = (\theta_{aff}, \theta_v)$. The affine component is the composition of three rotations ($\theta_{rot}$), three scalings ($\theta_{sc}$), three shears ($\theta_{sh}$), and three translations ($\theta_{tr}$). All these parameters are independently sampled from continuous uniform distributions with predefined ranges: $\mathcal{U}(a_{rot}, b_{rot})$, $\mathcal{U}(a_{sc}, b_{sc})$, $\mathcal{U}(a_{sh}, b_{sh})$, and $\mathcal{U}(a_{tr}, b_{tr})$, respectively. The deformable component is a diffeomorphic transform, obtained by integrating a smooth, random stationary velocity field (SVF) with a scaling and squaring approach (Moler and Van Loan, 2003; Arsigny et al., 2006), implemented efficiently for a GPU (Dalca et al., 2019b; Krebs et al., 2019). The SVF is generated by first sampling the parameters $\theta_v$. This is a random, low-resolution tensor (size $c_v \times c_v \times c_v \times 3$), where each element is a sample from a zero-mean Gaussian distribution with standard deviation $\sigma_{svf}$. This tensor is subsequently upscaled to the desired image resolution with trilinear interpolation, to obtain a smooth SVF, which is integrated to obtain $\phi_v$. The final deformed label map is obtained by resampling

$$L = S_i \circ \phi = S_i \circ [\phi_{aff}(\theta_{aff}) \circ \phi_v(\theta_v)] \tag{3}$$

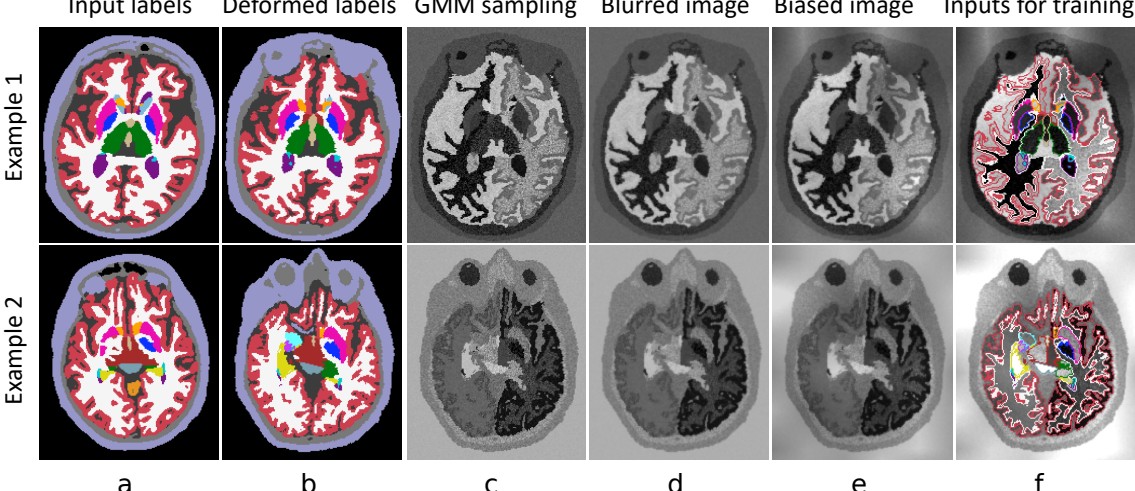

Figure 2: Intermediate steps of image generation (axial slices of 3D volumes). (a) Segmentation. (b) Warp with random smooth deformation field. (c) Image intensities sampled via a GMM with random parameters. (d) Blur. (e) Random bias field. (f) Synthesized images with the contours of the corresponding label maps.

with nearest neighbor interpolation. This generative model yields a wide distribution of neuroanatomical shapes, while ensuring spatial smoothness (Figure 2b).

Given the segmentation $L$, we sample a synthetic image $I$ as follows. First, we sample an image $G$ conditioned on $L$, following the likelihood model introduced in section 2.1, one voxel at the time using $G_j \sim \mathcal{N}(\mu_{L_j}, \sigma_{L_j}^2)$. The Gaussian parameters $\{\mu_k, \sigma_k\}$ are a set of $K$ independent means and standard deviations drawn from continuous uniform distributions $\mathcal{U}(a_\mu, b_\mu)$ and $\mathcal{U}(a_\sigma, b_\sigma)$, respectively. Sampling independently from a wide range of values yields images of extremely diverse contrasts (Figure 2c). To mimic partial volume effects, we make the synthetic images more realistic by introducing a small degree of spatial correlation between neighboring voxels. This is achieved by blurring $G$ with a Gaussian kernel $R(\sigma_{blur})$ with standard deviation $\sigma_{blur}$ voxels, i.e., $G^{blur} = G * R(\sigma_{blur})$ (Figure 2d).

We corrupt the images with a bias field $B$, parameterized by $\theta_B$. $B$ is generated in a similar way as the SVF: $\theta_B$ is a random, low resolution tensor (size $c_B \times c_B \times c_B$ in our experiments), whose elements are independent samples of a Gaussian distribution $\mathcal{N}(0, \sigma_b)$. This tensor is upscaled to the image size of $L$ with trilinear interpolation, and the voxel-wise exponential is taken to ensure non-negativity. The bias field corrupted image $G^{bias}$ is obtained by voxel-wise multiplication: $G^{bias} = G^{blur} \odot B$ (Figure 2e).

Finally, the training image $I$ is generated by standard gamma augmentation and normalization of intensities. We first sample $\gamma$ from a uniform distribution $\mathcal{U}(a_\gamma, b_\gamma)$ and then:

$$I_j = \left( [G_j^{bias} - \min_j(G_j^{bias})] \Big/ [\max_j(G_j^{bias}) - \min_j(G_j^{bias})] \right)^\gamma. \tag{4}$$

Table 1: Hyperparameters used in our experiments. Angles are in degrees; spatial measures are in voxels. Intensity hyperparameters assume an input in the [0,255] interval.

| $a_{rot}$ | $b_{rot}$ | $a_{sc}$ | $b_{sc}$ | $a_{sh}$ | $b_{sh}$ | $a_{tr}$ | $b_{tr}$ | $\sigma_{svf}$ | $a_\mu$ | $b_\mu$ | $a_\sigma$ | $b_\sigma$ | $\sigma_{blur}$ | $\sigma_b$ | $a_\gamma$ | $b_\gamma$ | $c_v$ | $c_B$ |
|---|---|---|---|---|---|---|---|---|---|---|---|---|---|---|---|---|---|---|
| -10 | 10 | 0.9 | 1.1 | -0.01 | 0.01 | -20 | 20 | 3 | 25 | 225 | 5 | 25 | 0.3 | 0.5 | -0.3 | 0.3 | 10 | 4 |

**Training:** Starting from a set of label maps, we use the generative process described above to form training pairs $\{I, L\}$ (Figure 2f). These pairs – each sampled with different parameters – are used to train the CNN in a standard supervised fashion (Figure 1).

### 2.3. Implementation details

**Architecture:** We use a U-Net style architecture (Ronneberger et al., 2015; Çiçek et al., 2016) with 5 levels of 2 layers each. The first layer contains 24 feature maps, and this number is doubled after each max-pooling, and halved after each upsampling. Convolutions are performed with kernels of size $3 \times 3 \times 3$, and use the Exponential Linear Unit as activation function (Clevert et al., 2016). We also make use of batch-normalization layers before each max-pooling and upsampling layer (Ioffe and Szegedy, 2015). The last layer uses a softmax activation function. The loss function is the average soft Dice (Milletari et al., 2016) coefficient between the ground truth segmentation and the probability map corresponding to the predicted output.

**Parametric distributions and intensity constraints:** The proposed generative model involves several hyperparameters (described above), which control the priors of model parameters. In order to achieve invariance to input contrast, we sample the hyperparameters of the GMM (describing priors for intensity means and variances) from wide ranges in an independent fashion, generally leading to unrealistic images (Figure 2). The deformation hyperparameters are chosen to yield a wide range of shapes – well beyond plausible anatomy. We emphasize that the hyperparameter values, summarized in Table 1, are *not* chosen to mimic a particular imaging modality or subject cohort.

**Skull stripping:** The proposed method is designed to segment brain MRI without any preprocessing. However, in practice, some brain MRI datasets do not include extracerebral tissue, for example due to privacy issues. We build robustness to skull-stripped images into our method, by treating all extracerebral regions as background in 20% of training samples.

**GPU implementation:** Our model, including the image sampling process, is implemented on the GPU in Keras (Chollet, 2015) with a Tensorflow backend (Abadi et al., 2016).

### 3. Experiments and results

We provide experiments to evaluate segmentation of *unprocessed* scans, eliminating the dependence on additional tools which can be CPU intensive and require manual tuning.

### 3.1. Datasets

We use four datasets with an array of modalities, and contrast variations within modalities. All datasets contain labels for 37 regions of interest (ROIs), with the same labeling protocol.

**T1-39:** 39 whole head T1 scans with manual segmentations (Fischl, 2012). We split the dataset into subsets of 20 and 19 scans. We use the labels maps of the first 20 as the only inputs to train SynthSeg, and evaluate on the held-out 19. We augmented the manual labels with approximate segmentations for skull, eye fluid, and other extra-cerebral tissue, computed semi-automatically with in-house tools, to enable synthesis of full head scans.

**T1mix:** 1,000 T1 whole head MRI scans collected from seven public datasets: ABIDE (Di Martino et al., 2014), ADHD200 (Consortium, 2012), GSP (Holmes et al., 2015), HABS (Dagley et al., 2017), MCIC (Gollub et al., 2013), OASIS (Marcus et al., 2007), and PPMI (Marek et al., 2011). Although these scans share the same modality, they exhibit variability in intensity distributions and head positioning due to differences in acquisition platforms and sequences. Since manual delineations are not available for these scans, we evaluate against automated segmentations obtained with FreeSurfer (Fischl, 2012; Dalca et al., 2018). T1mix enables evaluation on a large dataset of heterogeneous T1 contrasts.

**FSM:** 18 subjects, each with 3 modalities: T1, T2, and a sequence typically used in deep brain stimulation (DBS). The DBS scan is an MP-RAGE with: TR = 3000 ms, TI = 406 ms, TE = 3.56 ms, $\alpha = 8°$. With no manual delineations available, for evaluation we use automated segmentations produced by FreeSurfer on the T1 channel as ground truth for all modalities. This dataset enables evaluation on two new contrasts, T2 and DBS.

**T1-PD-8:** T1 and proton density (PD) scans for 8 subjects, with manual delineations. These scans were approximately skull stripped prior to availability. Despite its smaller size, this dataset enables evaluation on another contrast (PD) that is very different than T1.

Although FreeSurfer segmentations are not as accurate as manual delineations, they enable evaluation where manual labels are missing. FreeSurfer has been thoroughly evaluated on numerous independent datasets (Fischl et al., 2002; Tae et al., 2008). It also yields high Dice scores against manual segmentations for T1-39 (0.88, albeit biased by mixing FreeSurfer training and testing data) and T1-PD-8 (0.85).

### 3.2. Competing methods

We compare our method SynthSeg with three other approaches:

**Fully supervised network:** We train a *supervised* U-Net on the 20 training images from the T1-39 dataset (whole brain, unprocessed), aiming to assess difference in performance when testing on images of the same contrast (T1) acquired on the same and other platforms. We employ the same architecture and loss function as for SynthSeg, and we use the same data augmentation when applicable, specifically spatial deformation, gamma augmentation, and normalization of intensities. This supervised network can only segment T1 scans, so we refer to it as "T1 baseline".

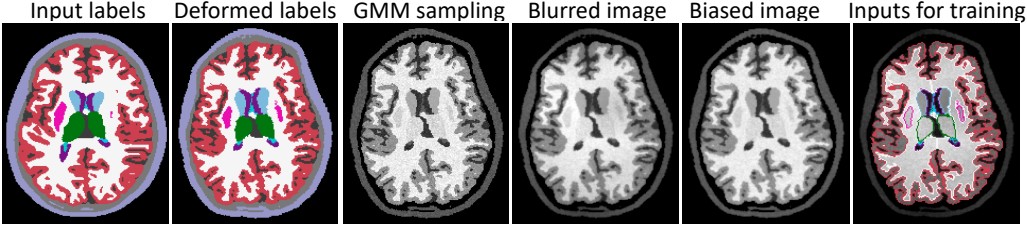

Figure 3: Generation of a T1-like image for training SynthSeg-rule.

**SAMSEG:** Based on the traditional Bayesian segmentation framework, SAMSEG (Puonti et al., 2016) uses unsupervised likelihood distributions, and is thus fully contrast-adaptive. Like our method, SAMSEG can segment both unprocessed or skull-stripped scans. SAMSEG does not rely on neural networks, and thus does not require training, but instead employs an independent optimization for each scan requiring tens of minutes.

**SynthSeg-rule:** We also analyze a variant of our proposed method, where the intensity parameters are representative of the test scans to be segmented. For each of the seven contrasts present in the training data (T2, PD, DSB, and four varieties of T1), we build a Gaussian hyperprior for the means and standard deviations of each label, using ground truth segmentations. At training, for every mini-batch we sample one of the seven contrasts, then we sample the means and standard deviations for each class conditioned on the contrast. This variant enables us to compare the generation of unrealistic contrasts during training, against enforcing prior information on the target modalities, if available. An example of these more realistic synthetic images (conditioned on T1 contrast) is shown in Figure 3.

### 3.3. Experimental setup

All CNN methods are trained on the training subset of T1-39, with our method variants only requiring the segmentation maps, whereas the supervised baseline also uses the T1 scans. We evaluate all approaches on the test subset of T1-39, as well as all of T1mix, T1-PD-8, and FSM. The T1 baseline is not tested on modalities other than T1, nor on T1-PD-8 because it cannot cope with skull stripped data. We assess performance with Dice scores, computed for a representative subset of 12 brain ROIs: cerebral white matter (WM) and cortex (CT), lateral ventricle (LV), cerebellar white matter (CW) and cortex (CC), thalamus (TH), caudate (CA); putamen (PU), pallidum (PA), brainstem (BS), hippocampus (HP), and amygdala (AM). We averaged results for contralateral structures.

### 3.4. Results

Table 2 provides a summary of the methods and their runtime. Figure 4 shows box plots for each ROI, method, and dataset, as well as averages across the ROIs. Table 3 shows corresponding median scores and p values using Wilcoxon test. Finally, Figure 5 shows sample segmentations for every method and dataset. The supervised T1 baseline excels when tested on the test scans of T1-39 (i.e., intra-dataset), achieving a mean Dice of 0.89, and outperforming all the other methods for every ROI. However, when tested on T1 images

Table 2: Summary of results, capturing the performance of each method, its ability to segment arbitrary modalities, and run time (averaged over 10 runs). SAMSEG was run using 8 cores (Intel Xeon at 3.00GHz), whereas SynthSeg was run on an Nvidia P6000 GPU. Image loading time was not considered.

| Method | Overall performance | modality-agnostic | runtime (s) |
|---|---|---|---|
| Supervised | 0.89 ± 0.10 (same dataset) 0.59 ± 0.11 (other T1s) | No | 3.06 ± 0.02 |
| SAMSEG | 0.83 ± 0.02 | Yes | 1382 ± 192 |
| SynthSeg-rule | 0.82 ± 0.02 | Yes | 3.22 ± 0.03 |
| SynthSeg | 0.85 ± 0.02 | Yes | 3.22 ± 0.03 |

Table 3: Median Dice scores and p values for two-sided non-parametric Wilcoxon signed-rank tests comparing SynthSeg and the competing methods. Sub-datasets of T1mix are marked with a star.

| Dataset | SynthSeg Med. Dice | T1-baseline Med. Dice | T1-baseline p value | SAMSEG Med. Dice | SAMSEG p value | SynthSeg-rule Med. Dice | SynthSeg-rule p value |
|---|---|---|---|---|---|---|---|
| T1-39 | 0.861 | 0.894 | $p < 10^{-3}$ | 0.849 | $p < 10^{-3}$ | 0.819 | $p < 10^{-3}$ |
| T1mix | 0.852 | 0.601 | $p < 10^{-94}$ | 0.858 | $p < 10^{-30}$ | 0.806 | $p < 10^{-85}$ |
| ABIDE* | 0.838 | 0.761 | $p < 10^{-15}$ | 0.856 | $p < 10^{-11}$ | 0.799 | $p < 10^{-19}$ |
| ADHD* | 0.843 | 0.649 | $p < 10^{-14}$ | 0.857 | $p < 10^{-4}$ | 0.804 | $p < 10^{-9}$ |
| HABS* | 0.858 | 0.630 | $p < 10^{-4}$ | 0.859 | 0.7 | 0.819 | $p < 10^{-4}$ |
| GSP* | 0.853 | 0.549 | $p < 10^{-92}$ | 0.857 | $p < 10^{-7}$ | 0.806 | $p < 10^{-90}$ |
| MCIC* | 0.869 | 0.750 | $p < 10^{-5}$ | 0.863 | $6.4 \times 10^{-3}$ | 0.829 | $p < 10^{-5}$ |
| OASIS* | 0.855 | 0.857 | $p < 10^{-4}$ | 0.867 | $p < 10^{-10}$ | 0.812 | $p < 10^{-12}$ |
| PPMI* | 0.851 | 0.726 | $p < 10^{-12}$ | 0.861 | $p < 10^{-6}$ | 0.811 | $p < 10^{-12}$ |
| T1 FSM | 0.869 | 0.531 | $p < 10^{-13}$ | 0.869 | 0.6 | 0.827 | $p < 10^{-3}$ |
| T2 FSM | 0.841 | N/A | N/A | 0.822 | $1.2 \times 10^{-3}$ | 0.822 | $1.5 \times 10^{-2}$ |
| DBS FSM | 0.828 | N/A | N/A | 0.821 | $3.8 \times 10^{-3}$ | 0.831 | 0.2 |
| T1-PD8 | 0.848 | N/A | N/A | 0.823 | $1.7 \times 10^{-2}$ | 0.810 | $1.2 \times 10^{-2}$ |
| PD-PD8 | 0.830 | N/A | N/A | 0.801 | $3.6 \times 10^{-2}$ | 0.830 | 0.4 |

from T1mix and FSM, we observe substantial variations in Dice scores (e.g., across the different sub-datasets within T1mix), with a consistent decrease in performance (see for instance the segmentation of the T1 in FSM in Figure 5). This is likely due to the limited variability in the training dataset, despite the use of augmentation techniques, highlighting the challenge of variation in *unprocessed* scans from different sources, even within the same modality.

SAMSEG yields very uniform results across datasets of T1 contrasts, producing mean Dice scores within 3 points of each other. Being agnostic to contrast, it outperforms the T1 baseline outside its training domain. It also performs well for the non-T1 contrasts. Although the mean Dice scores are slightly lower than for the T1 datasets (which nor-

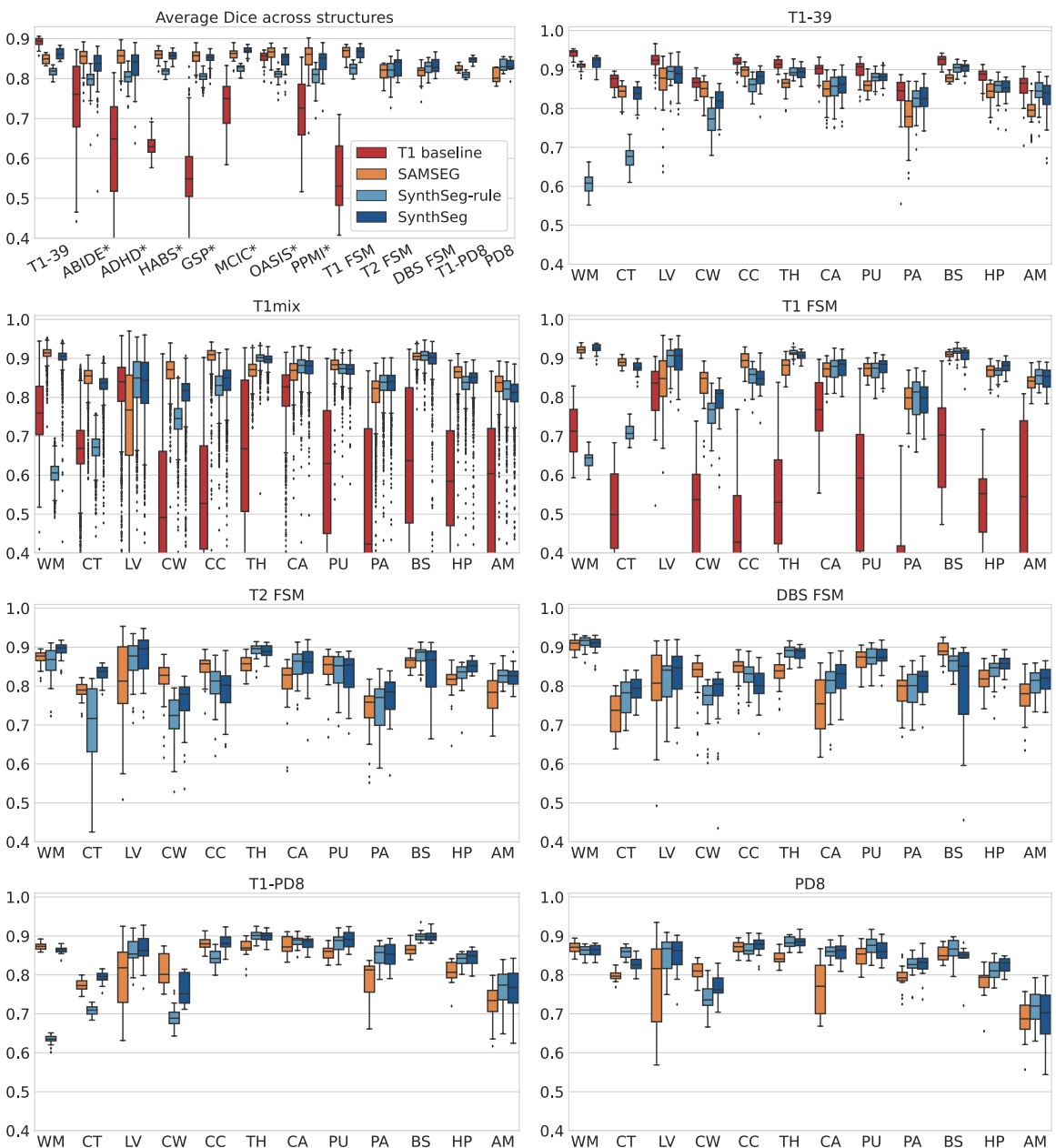

Figure 4: Dice scores obtained by each method, shown both in aggregate (top left), and for individual ROIs on each dataset. Sub-datasets of T1mix are marked with a star.

mally display better contrast between gray and white matter), they remain robust for every contrast and dataset with minimum mean Dice of 0.81.

SynthSeg also produces high Dice across all contrasts, slightly higher than SAMSEG (0.02 mean Dice improvement), while requiring a fraction of its runtime (Table 2). The

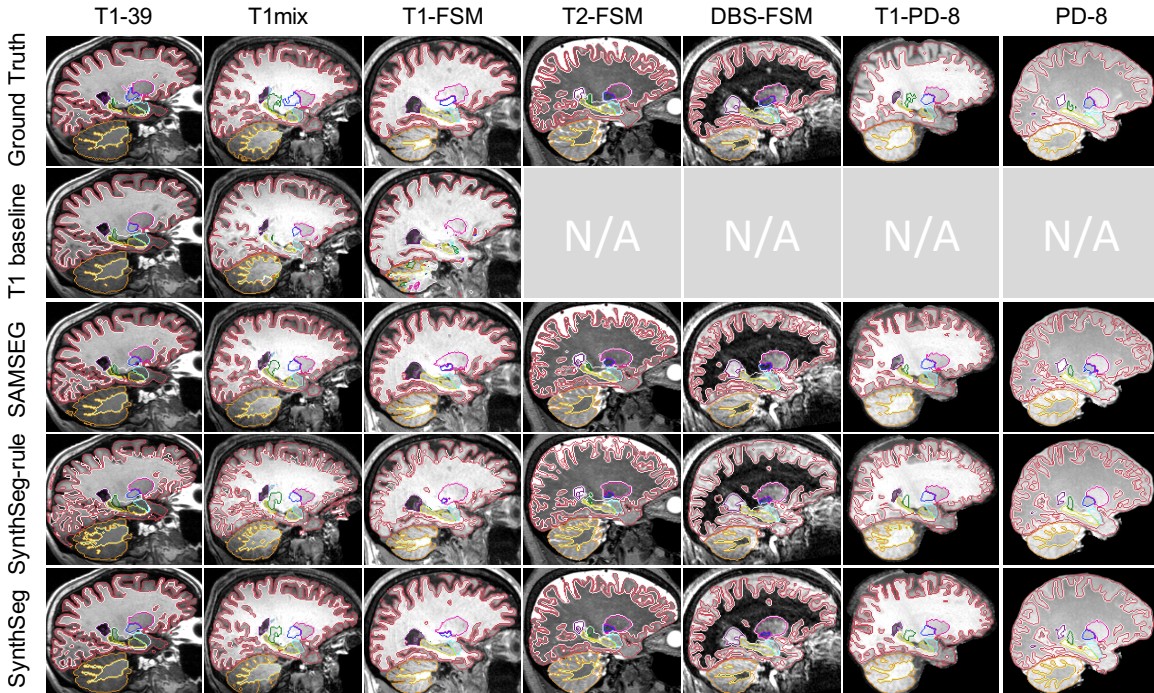

Figure 5: Example segmentations for each method and dataset. We selected the median subject in terms of Dice scores across ROIs and methods.

difference between SAMSEG and SynthSeg is smaller for T1mix and T1 FSM because SAMSEG is positively biased by the use of FreeSurfer segmentations as ground truth, since the methods work similarly. The improvement of SynthSeg compared to SAMSEG is consistent across structures, except the cerebellum. Compared to the T1 baseline, the mean Dice is 0.03 lower on the supervised training domain (T1-39), but generalizes significantly better to other T1 datasets, and can segment other MRI contrasts with little decrease in performance (minimum mean Dice is 0.83).

Importantly, SynthSeg-rule is outperformed by SynthSeg, and its Dice scores are also slightly lower than those produced by SAMSEG. This illustrates that adapting the parameters to a certain contrast is counterproductive, at least within our simple generative model: we observe consistent drops in performance across ROIs and datasets, despite injecting contrast-specific knowledge for each modality. This result is consistent with recent results in image augmentation (Chaitanya et al., 2019), and supports the theory that forcing the network to learn to segment a broader range of images than it will typically observe at test time improves generalization.

## 4. Discussion and conclusion

We presented a learning strategy for modality-agnostic brain MRI segmentation, which builds on classical generative models for Bayesian segmentation. Sampling a wide range

of model parameters enables the network to learn to segment a wide variety of contrasts and shapes during training. At test time, the network can therefore segment neuroanatomy given an unprocessed scan of any contrast in seconds. While the network is trained in a supervised fashion, the only data required are a few label maps. Importantly, we do not require any real scans during training, since images are synthesized from the labels, and are thus always perfectly aligned – in contrast to techniques relying on manual delineations. Our method requires the training label maps to contain labels for all brain structures to be synthesized.

While a supervised network excels on test data from the same domain it was trained on, its performance quickly decays when faced with more variability, even within the same type of MRI contrast. We emphasize that this effect is particularly pronounced as we tackle the challenging task of segmentation starting with *unprocessed* scans. This is one reason why deep learning segmentation techniques have not yet been adopted by widespread neuroimaging packages like FreeSurfer or FSL, where fewer assumptions on the specific MRI contrast of the user's data need to be made. In contrast, SynthSeg maintains accuracy across T1 variants as well as other MRI modalities.

In absolute terms, SynthSeg's Dice scores are consistently high: higher than SAMSEG, and not far from *supervised* contrast-specific networks, like the T1 baseline or scores reported in recent literature (Roy et al., 2019). Compared with our recent article that uses a CNN to estimate the GMM and registration parameters of the Bayesian segmentation framework (Dalca et al., 2019a), the method proposed here achieves higher average Dice on T1 (0.86 vs 0.82) and PD datasets (0.83 vs 0.80). However, we highlight that direct comparison is not available due to differences in datasets: in this work, we could only use 19 subjects from T1-39 for evaluation. More importantly, our previous method requires significant preprocessing and modality-specific unsupervised re-training. This highlights the ability of our new method to segment a wide variety of contrasts without retraining or preprocessing; the latter eliminates the dependence on additional tools which can be computationally expensive and require manual tuning.

We believe that the proposed learning strategy is applicable to many generative models from which sampling can yields sensible data, even beyond neuroimaging. By greatly increasing the robustness of fast segmentation CNNs to a wide variety of MRI contrast, without any need for retraining, SynthSeg promises to enable adoption of deep learning segmentation techniques by the neuroimaging community.

## Acknowledgments

This research was supported by the European Research Council (ERC Starting Grant 677697, project BUNGEE-TOOLS), and by the EPSRC-funded UCL Centre for Doctoral Training in Medical Imaging (EP/L016478/1) and the Department of Health's NIHR-funded Biomedical Research Centre at University College London Hospitals. Further support was provided in part by the BRAIN Initiative Cell Census Network grant U01-MH117023, the National Institute for Biomedical Imaging and Bioengineering (P41-EB015896, 1R01-EB023281, R01-EB006758, R21-EB018907, R01-EB019956), the National Institute on Aging (1R56-AG064027, 5R01-AG008122, R01-AG016495, 1R01-AG064027), the National Institute of Mental Health the National Institute of Diabetes and Digestive and Kidney Dis-

eases (1-R21-DK-108277-01), the National Institute for Neurological Disorders and Stroke (R01-NS112161, R01-NS0525851, R21-NS072652, R01-NS070963, R01-NS083534, 5U01-NS086625,5U24-NS10059103, R01-NS105820), and was made possible by the resources provided by Shared Instrumentation Grants 1S10-RR023401, 1S10-RR01-9307, and 1S10-RR023043. Additional support was provided by the NIH Blueprint for Neuroscience Research (5U01-MH093765), part of the multi-institutional Human Connectome Project. In addition, BF has a financial interest in CorticoMetrics, a company whose medical pursuits focus on brain imaging and measurement technologies. BF's interests were reviewed and are managed by Massachusetts General Hospital and Partners HealthCare in accordance with their conflict of interest policies.

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
