# OpenReview forum: "A learning strategy for contrast-agnostic MRI segmentation"
_MIDL.io/2020/Conference — MIDL 2020_

### Official Review · AnonReviewer4 · 2020-02-27
**Good paper tackling segmentation of brain structures in MR images of unseen contrasts**

**Rating:** 3
**Confidence:** 4
**Recommendation:** Oral, Poster

**Summary:**

The authors propose a training scheme to make CNNs for segmentation agnostic to MR contrasts unseen during training. A generative model creates synthetic MR images conditioned on label maps. These generated images together with the label maps are then used to train a segmentation network. This network is able to segment MR images of contrast never seen. The approach could be of high significance, for example, when training examples are scarce, when robustness to multi-centre data is required, or when new MR contrasts are introduced.

**Strengths:**

-	The paper is well structured and written, making it understandable and easy to follow
-	Strong motivation and introduction of existing literature
-	Well formulated methods section with algorithm and code for reproducibility
-	Promising results
-	Extensive evaluation: Two baselines and an ablation of the proposed AnyNET serve as comparison
-	Four different datasets were considered with label maps consisting of multiple ROIs


**Weaknesses:**

There are two very strong claims, which need to be weakened in my opinion:
-	“for the first time”: how about “to the best of our knowledge for the first time”
-	“not biased towards any”: any is a very strong statement, and hard to prove (if not impossible). How about removing the any? Same in the last paragraph of the introduction in italic.

I wonder what the influence of the number of labels in the label map $S_i$ is. This is not discussed in the paper. Also, could it be that the algorithm is heavily location-dependent by simply learning the position of the different brain structures (i.e., learning an atlas) instead of being contrast-agnostic? Did the author, for instance, try to translate the testing images (beyond the $\pm$ 20 px used during the image generation). Another experiment supporting your claims would be to use e.g. brain tumor images from the BraTS dataset (e.g., T1c image). If it would be truly contrast-agnostic, the contrast differences introduced by the tumor should not affect the segmentation very strongly (very strong is, of course, subjective by visual inspection since ground truth is not available and probably difficult to obtain with existing methods for the used ROIs).

**Detailed Comments:**

Fig 1 and Fig 2: can the authors add the input image without the contour overlay as a separate column (the contour overlay makes it difficult to see something).

I find the statement “nor on T1-PD-8 because it cannot cope with skull stripped data” unfair because you trained your method with skull-stripped data, which could have been done for the baseline too.

Minor:
-	Introduction: “has been hindered by the high variety in MRI contrasts”: is there any reference on this? E.g., how about literature on robustness to inter-/intra-scanner/site variations?
-	Introduction: the “unpreprocessed” is confusing in a sense that you argue about the 1 mm voxel size in the introduction but do not tackle this with your method. You are referring to pre-processing altering the intensity? Was there any intensity normalization for AnyNET during the testing?
-	Introduction: the statement “even within the same MRI contrast” is confusing
-	I prefer having T1w as an abbreviation for T1-weighted MR images (same for T2w).
-	Fig 3: data points are cut off for some datasets. Y-axis label vary between “Dice Scores” and “Dice”

Typos:
-	p3 last paragraph of Sec 2.1: algorihtm -> algorithm
-	p5 Training paragraph: forms -> form
-	p6: Reference for ADHD200 is abbreviated as noa, 2012
-	Section 4 title: conclusion is capitalized while the other section titles are lowercase
-	p10: comutationally -> computationally
-	Inconsistencies in pre-processing and preprocessing
-	Fig 3 caption: statistics capitalized


**Justification Of Rating:**

I think the paper makes a great contribution to MIDL 2020 and is of high relevance for the segmentation community. Few adaptions (see questions to address in the rebuttal) might convince me to increase the rating to strong accept.

**Paper Type:**

methodological development

**Questions To Address In The Rebuttal:**

I rate the paper as methodological. Therefore, I am happy if the authors weaken the strong claims. Please also fix the typos (detailed comments).

**Special Issue:**

no

---

> ### Author Response · Authors · 2020-03-27
> **Response to AnonReviewer4**
>
> We thank the reviewer for the very detailed feedback, and we will include all the minor comments. We also agree that the claims on being first to achieve contrast robustness and to be able to segment any modality are too strong, and we will soften them as suggested. We will also clarify that each training label map must contain labels for all structures to be generated, and segmented.
>
> We believe it is highly unlikely that the presented method blindly applies a learnt atlas to all test scans, as 1) none of the datasets were rigidly aligned beforehand, and 2) the position and orientation of test scans already fluctuate beyond the +/- 20 pixels maximum translation used during training.
>
> Finally, we thank the reviewer for the excellent suggestion about using this method to segment tumours. However, this would require a proper anatomical model of tumours, which is not encompassed in the current training label maps. We are excited to explore this direction for future work.

---

### Official Review · AnonReviewer3 · 2020-03-13
**contrast-agnostic semantic segmentation: very relevant topic, well written paper, some information and analyses missing**

**Rating:** 3
**Confidence:** 5
**Recommendation:** Poster

**Summary:**

The paper describes a deep learning strategy to allow for segmentation of brain MR scans regardless of the actual contrast of the MR datasets. Synthetic, simulated data was used for initial training. Evaluation was performed on in-vivo data of more than 1000 subjects. Results show that the proposed approach performs better than previous methods like the classical Bayesian segmentation.


**Strengths:**

- well structured paper, easy to read.
- paper targets a very relevant topic. Current work on MR image segmentation often works on one contrast the algorithm was trained on, but fails on new/other contrasts. Therefore, contrast-agnostic segmentation is a very important topic.
- the use of in-vivo data. The work evaluates on real world data and not only on simulated data.


**Weaknesses:**

- limited range of contrasts. The contrasts used (T1, T2, PD) are relatively close in appearance in healthy brains (in some aspects T1 and T2 show inverted intensities). It would have been clinical more interesting to include other contrast like DIR or diffusion-weighted imaging)
- It would be helpful to provide a table with contrast-relevant scan parameters for all data sources.
- no information is provided on the actual spatial resolution of the datasets. Are they identical? Has the data be reformatted to yield the same nominal resolution. This is important to be able to compare the results!
- no groundtruth (manual segmentation) available for by far the largest part of the in-vivo data.


**Justification Of Rating:**

This is relevant work and the results are encouraging. There is some missing data and information. In addition, the impact of the described work should be improved by providing more detailed and objective analyses.


**Paper Type:**

methodological development

**Questions To Address In The Rebuttal:**

I am wondering about the practical impact of this work, since a rather limited range of contrasts are considered. I fully agree with you that the dependency of segmentation results on different MR contrast is quite high and can produce wrong findings. Nonetheless, in practice, if segmentation of brain is needed (e.g. for atrophy measurements) a dedicated sequence (usually T1-weighted MP-RAGE) is used. This is more or less standard, although there might be a range of sequence varieties producing slightly different T1-contrasts. I think, the analysis you should provide is not (only) between T1 and T2 variants, but also a more detailed analysis on how valuable your approach is for variations in T1-contrast. This would mean that you should analyse your sub-group T1mix separately and split it up in the different T1-contrast variants.

A drawback is that you do not have ground truth data for most of your data, except for the FreeSurfer result. Did you compare FreeSurfer results with manual segmentation for the cases, which had the manual segmentation groundtruth?

The results of SAMSEG for T1mix and T1 FSM are significantly better than for the other datasets. This is a bias in your analysis, since you are creating groundtruth segmentation with FreeSurfer for those two datasets. However, FreeSurfer uses a very similar algorithm (Bayesian segmentation) as SAMSEG. It would provide a better reference if you would also use FreeSurfer groundtruth for the other (T1-weighted) data.

**Special Issue:**

no

---

> ### Author Response · Authors · 2020-03-27
> **Response to AnonReviewer3**
>
> We thank the reviewer for the insights and ideas.
>
> We agree that adding more MRI contrasts, especially clinical, to the four already presented (T1, T2, PD, and DBS) would help broadening the scope of this study. This will certainly be added in future work. However, we also believe that experiments on these four contrasts indicate the promise of the method.
>
> We also agree that studying the variations of segmentation accuracies within the heterogeneous T1mix dataset is important, beyond comparing to the T1 modalities from the other datasets. We conducted this experiment and we will add a box plot to Figure 3 of the camera-ready version. The results show that SAMSEG and AnyNet sustain Dice scores above 0.8 across T1mix sub-datasets, while the T1-baseline yields highly fluctuating results, with medians ranging from 0.55 to 0.85.
>
> We agree that using FreeSurfer (FS) labels as ground truth is not ideal, but we note that FS has been thoroughly evaluated on numerous independent datasets [1-3]. Hence, we believe that FS segmentations provide a compromise to enable evaluation on very large datasets such as T1mix, where it is impossible to obtain manual segmentations. We will include the average Dice score between FS and manual segmentations (0.88 for T1-39, albeit biased by mixing FS training and test data, and 0.85 for T1-PD-8) to support this position.
>
> We also agree that the SAMSEG baseline is positively biased when using FreeSurfer segmentations as ground truth. However, this is not an issue in our study, since the bias works against our proposed method. We will mention it in the manuscript.
>
> Even if isotropic T1s scans are common in neuroimaging research, we think that our method can have a big impact in many settings, such as clinical acquisition protocols, where such isotropic T1w scans are often missing.
>
>
> [1] Fischl, B., Salat, D. H., Busa, E., Albert, M., Dieterich, M., Haselgrove, C., ... & Montillo, A. (2002). Whole brain segmentation: automated labeling of neuroanatomical structures in the human brain. Neuron, 33(3), 341-355.
>
> [2] Han, X., & Fischl, B. (2007). Atlas renormalization for improved brain MR image segmentation across scanner platforms. IEEE transactions on medical imaging, 26(4), 479-486.
>
> [3] Tae, W. S., Kim, S. S., Lee, K. U., Nam, E. C., & Kim, K. W. (2008). Validation of hippocampal volumes measured using a manual method and two automated methods (FreeSurfer and IBASPM) in chronic major depressive disorder. Neuroradiology, 50(7), 569.

---

### Official Review · AnonReviewer1 · 2020-03-17
**Interesting approach to use generative models to train contrast agnostic deep models**

**Rating:** 3
**Confidence:** 3
**Recommendation:** Poster

**Summary:**

In this paper "A learning strategy for contrast-agnostic MRI segmentation", the authors tackle the problem of image segmentation on MRI data of varying contrast. They show that using a generative model (deformation, bias field, intensity model with randomized parameters) leads to a training approach that creates networks that are not sensitive to the effects over which the network is trained.

**Strengths:**

The method is well motivated both from a practical perspective and from a theoretical one. The technical description is well considered and clearly presented. The literature is summarized well. On the target metrics, the method is highly effective.

**Weaknesses:**

The biggest weakness of this approach is that it is based off a core network that compares against baselines that are under trained relative to current state of the art. For example, a careful comparison of modality specific T1 tools would be very helpful. Considerations of variance / statistical tests were not performed even though the data were well visualized for variability.

**Justification Of Rating:**

This is an interesting well written paper. The method is articulated clearly in terms of both motivation and technical details. The details are sufficient such that others could reproduce this work. Two areas of improvement would be to ensure that a robust modern baseline is included and include a statistical assessment.

**Paper Type:**

methodological development

**Questions To Address In The Rebuttal:**

Statistical assessments should be included in the review and it would be helpful to connect the intermodality performance for what a well trained single modality approach would achieve.

**Special Issue:**

no

---

> ### Author Response · Authors · 2020-03-27
> **response to AnonReviewer1**
>
> We thank the reviewer for suggesting the inclusion of statistical assessment. We will report results of two-sided non-parametric Wilcoxon signed-rank tests between AnyNET and the competing methods. For instance, p-values between AnyNet and SAMSEG for T1-FSM, T2-FSM, and DBS-FSM are respectively 0.59 (i.e., no significant difference), 1.18e-3, and 3.78e-3.
>
> Regarding the undertrained T1-baseline, we are not exactly sure of what the reviewer meant, but we clarify the following related points: 1) this work focuses on the sampling strategy, which is applicable for any segmentation architecture, and hence the exact architecture design was not our focus, 2) we believe the UNet architecture is still a representative network for segmentation often used in many medical imaging applications, 3) robustness is often improved with data augmentation, which we used thoroughly for all networks, including the supervised baseline. We agree that using a better T1-baseline could help reduce the domain gap across T1 datasets, and we will add this point to the discussion. However, we believe the main conclusions of this study unaffected: AnyNet is able to segment highly varying MRI contrasts, which cannot be achieved with other segmentation networks, and AnyNet outperforms SAMSEG while running considerably faster.

---

### Official Review · AnonReviewer2 · 2020-03-19
**Clear accept: Novel learning strategy for contrast-agnostic Brain MRI segmentation + excellent results**

**Rating:** 4
**Confidence:** 5
**Recommendation:** Oral

**Summary:**

Starting with a set of segmentation maps for Brain MRIs, a generative model is used to create a training dataset with simulated images of widely varying contrasts between tissue types. A segmentation CNN trained on this dataset performs well on real images acquired using different MRI protocols and from different datasets / imaging centers. In my opinion, this approach seems to essentially solve the cross-scanner / cross-protocol robustness problem in CNN-based Brain MRI segmentation.

**Strengths:**

* The paper tackles an important problem - that of lack of robustness of CNN-based image segmentation methods to scanner / protocol changes and proposes a simple, but effective solution for the same.

* The method can be seen as extensive data augmentation, except that it augments a dataset with no real MR images to begin with - all the training images are simulated. I think this is a neat solution to circumvent reliance on training images of a particular contrast.

* By including images with and without skull stripping, as well as by modeling the bias fields, the method ensures that test images can be segmented without any pre-processing. Although removing bias fields with other tools is relatively easy, it is nevertheless nice to have a segmentation tool that can work directly with the acquired images.

* The paper is extremely well-written and a pleasure to read!

**Weaknesses:**

There are no major weaknesses in the paper. Here are a few minor suggestions:

* Some details regarding the training procedure and CNN architecture would be helpful:
        - Does it super long for training the CNN, given the wide ranges of different hyper-parameters and therefore, the relatively large size of the effective training dataset. Although not relevant for practical use, I think this information might be interesting for readers.
        - I suppose the architecture does not contain any batch normalization layers. It would be better to clarify this, as BN layers are typically present in segmentation CNNs.
        - Do the authors observe any training instability due to the high variance in simulated contrasts?

* While extremely elegant, I think the method would be restricted to cases where the intensities within each segmentation label can be modeled as samples of a Gaussian distribution. For instance, suppose we are only interested in segmenting a structure in a particular region of the image and have labels for this, but the rest of image is just labelled as background although it might contain several other structures. In this situation, the proposed method to start with segmentation labels and generate training images would not work. For completeness, it would be nice if the authors can point this out in the discussion.

**Justification Of Rating:**

I believe that the paper tackles an important problem in Brain MRI segmentation and proposes a novel, elegant and effective solution for the same. This will certainly be of great interest to the community.

**Paper Type:**

both

**Special Issue:**

yes

---

> ### Author Response · Authors · 2020-03-27
> **Response for AnonReviewer2**
>
> We thank the reviewer for pointing that our method requires labels for all structures to be generated and segmented. We will add this clarification to our method. We will also specify that Batch Normalization layers are used before each max-pooling and up-sampling operation.

---

### Meta-Review · Area_Chair1 · 2020-04-06
**MetaReview of Paper72 by AreaChair1**

**Rating:** 4
**Recommendation For Accepted Papers:** Oral, Poster

**Metareview:**

A well-defined method for dealing with limited amount of ground truth (fully labeled) segmentation data which can be seen as a form of data augmentation. Validation on relatively close modalities weakens validity of the general claim that this method is entirely "contrast agnostic". For such a claim significantly larger variation of modalities  should be studied (as admitted by the authors). It is not obvious if entirely "contrast agnostic" (modality independent) features exist as this implies that NN should learn the structure (anatomy), which current NN are not know to be good at. While the paper shows some promise for limited data variability, it is not obvious how this approach would scale. Perhaps, the authors can use their own generative framework to synthesize unseen modalities to validate their claim.

From a practical point of view, the authors are strongly encouraged to include a more detailed analysis of T1 (as pointed out by R3). The authors are strongly encouraged to weaken their "the first time" claim.

**Paper Type:**

both

**Special Issue:**

no

---

> ### Author Response · Authors · 2020-05-11
> **Thanks + clarification**
>
> We thank the area chair for their summary and comments.
>
> We emphasize that, while algorithm performance may vary across modalities in practice, our method is not exposed to *any* real data (or statistics derived from real data) during training. Therefore, the learned features are driven to capture shape while being agnostic to contrast. This is in fact the main contribution of the manuscript: a network to segment brain structures on any MRI modality without retraining.  We demonstrate this on several distinct modalities in our experiments. We also emphasize that the synthetic samples mentioned by the area chair are exactly what we use in training, since every minibatch uses a different synthetic contrast for each structure.
>
> We will be happy to add a more detailed analysis of T1 (as shown in our response to R3) and weaken the wording of the claim as suggested.

---

### Decision · Program_Chairs · 2020-04-11

Accept